# Nutritional Habits of Professional Cyclists during Pre-Season

**DOI:** 10.3390/nu14183695

**Published:** 2022-09-07

**Authors:** José Joaquín Muros, Cristóbal Sánchez-Muñoz, Daniel Campos, Daniel Hinojosa-Nogueira, José Ángel Rufián-Henares, Manuel Mateo-March, Mikel Zabala

**Affiliations:** 1Department of Didactics of Corporal Expression, School of Education, University of Granada, 18011 Granada, Spain; 2Department of Physical Education and Sport, Faculty of Sport Sciences, University of Granada, 18011 Granada, Spain; 3Department of Paediatrics, School of Medicine, University of Granada, Avda. Investigación 11, 18016 Granada, Spain; 4Department of Nutrition and Food Science, Institute of Nutrition and Food Technology, Biomedical Research Centre, University of Granada, 18071 Granada, Spain; 5Sport Science Department, Universidad Miguel Hernández de Elche, 03202 Elche, Spain; 6Faculty of Sport Sciences, Universidad Europea de Madrid, 28670 Madrid, Spain

**Keywords:** nutrition, performance, sport, eating habits, cyclists

## Abstract

The first aim of the present study was to assess the dietary intake of professional cyclists during pre-season. The second aim was to assess the dietary habits of this population during a complete season. Fifteen elite male (age: 23.2 ± 5.4 years) and twenty-three elite female (age: 20.1 ± 7.0 years) cyclists volunteered to participate in the study. Dietary nutrient intake during pre-season was assessed using a 72 h dietary recall interview, and a 136-item food frequency questionnaire was used to assess dietary habits during the year. Protein intake exceeded the PRI’s recommendation of 0.83 g/kg/day for all cycling groups. Fat exceeded RI recommendations in females in both road (43.3%) and CXO (39.8%) cycling groups, whilst males were found to follow recommendations for fat intake. CHO intake was below recommendations in all groups. Intake of all vitamins exceeded recommendations, with the exception of B9 in female road cyclists (77.8% RDA) and vitamin D in all groups. With regards to mineral intake, consumption exceeded RDA/AI recommendations in all groups except for iodine in male XCO cyclists (61.6%), female road cyclists (61.6%), and female XCO cyclists (58%) and potassium in female road cyclists (74.6%). Males consumed greater amounts of eggs and non-processed foods than females. Road cyclists consumed greater amounts of fish and seafood and had a lower intake of coffee and tea than XCO cyclists. Better knowledge of food guidelines in terms of serving and food variety is important for professional cyclists at may impact health and performance.

## 1. Introduction

A typical pre-season in cycling lasts from November to January. During this period, cyclists focus on optimal recovery from the season and maintaining an optimal body composition. The nutritional intake of elite cyclists during this period should meet requirements in order to promote optimal recovery, maintain appropriate body composition, and provide essential micronutrients for health. While researchers have investigated nutrition during different cycling events [1,2,3,4,5], no data re currently available for eating patterns during a complete season or for the nutritional habits of professional cyclists during pre-season.

Carbohydrate (CHO) intake in cyclists is traditionally shown to be high during competition, as it is well-known that CHO ingestion helps to maintain blood glucose levels during exercise and replaces glycogen to prevent performance deterioration [6]. CHO intake during pre-season is also highly important in terms of daily recovery, fuelling light cycling training needs and supporting health [7]. It has been argued that CHO-rich food must provide the majority of energy in resistance sports such as cycling [8]. This higher CHO intake should be compensated by reducing protein intake although adequate protein intake is also important for regenerating tissue and avoiding muscle protein catabolism during and after training sessions [9]. Protein intake has been shown to exceed recommended daily intakes in endurance athletes [10]. High protein intake is related with urea production and oxidation of the carbon skeleton, whilst it also interferes with sufficient CHO intake [11]. Fat intake must also be adequate to supply essential fatty acids and liposoluble vitamins [10]. Fat consumption has also been shown to be excessive in endurance athletes [12]. It might be appropriate to reduce fat intake in order to make room for a greater CHO intake [13]. Greater consumption of micronutrients may be required due to increased needs for building, repairing, and maintaining lean body mass in athletes [14].

To the authors’ knowledge, no data evaluating the nutritional intake of elite cyclists during pre-season are available, with information on energy and nutrient requirements during this period also lacking. Thus, the first aim of the present study was to assess the dietary intake of professional cyclists during pre-season. The second aim was to assess dietary habits in this population during a complete season and highlight targets for intervention.

## 2. Materials and Methods

### 2.1. Participants

A cross-sectional study with fifteen elite male (age: 23.2 ± 5.4 years old, height: 181.2 ± 7.2 cm, weight: 69.6 ± 7.5; BMI: 21.1 ± 1.5 kg·m^−2^) and twenty-three elite female (age: 20.1 ± 7.0 years old, height: 164.8 ± 5.6 cm, weight: 58.2 ± 6.7; BMI: 21.3 ± 2.8 kg·m^−2^) national cyclists was conducted. The sample included European Championship, World Championship, and Olympic Games medallists, with participation being voluntary. Demographic data regarding specialty (road or cross-country Olympics (XCO)) are presented in Table 1. Cyclists were participating in a national pre-season training camp that took place in Alicante, Spain, between 7 November 2017 and 10 November 2019. Ethical principles of the Declaration of Helsinki for medical research were followed. Ethical approval was granted by the Ethics Committee of the University of Granada, Spain.

### 2.2. Dietary Assessment

Daily nutrient intake was assessed using a 72 h dietary recall interview considering intake over the previous three days, including one weekend day in order to capture weekly variations in weekdays and weekend. A well-trained professional nutritionist interviewed all participants in person. Participants were asked to recall all food consumed in the preceding 72 h, including supplements and beverages. Pictures of reference food portions were employed during the interview in order to improve the accuracy of reporting. Food records were analysed using the nutrient analysis program DIAL [15].

The 136-item food frequency questionnaire (FFQ) was used to assess dietary practices. This has demonstrated high validity and reliability within similar populations [16]. Commonly used portion sizes were specified (cup, teaspoon, etc.) for each food item. Participants were asked to report their average consumption of a specified unit of foodstuff over the previous year. Nine options were offered: never or hardly ever, one to three times a month, once a week, two to four times a week, five to six times a week, once a day, two to three times a day, four to six times a day, and more than six times a day. The selected response was converted into a daily intake measure (for example, responses of 5–6 times a week were converted to 0.78 servings per day). A seasonal variation factor was considered for foodstuffs whose consumption was not regular throughout the year.

A total of 19 food groups were defined (milk and dairy products, eggs, meat and meat products, fish and seafood, vegetables, pulses, fruits, nuts, pasta/potatoes/rice, refined cereals and their derivatives, wholegrain cereals and their derivatives, olive oil, other oils and fats, baked goods and pastries, sugar and sweets, beer/wine, alcoholic beverages, soft drinks, and coffee/tea) according to their nutritional similarities. Non-processed or minimally processed foods were classified as edible parts of plants (seeds, fruits, leaves, stems, and roots), animals (muscle, offal, eggs, and milk), and fungi or algae. Ultra-processed foods were classified as soft drinks, sweet or savoury packaged snacks, reconstituted meat products, and pre-prepared frozen dishes. Classifications were made in line with the groupings proposed by Monteiro et al. [17].

### 2.3. Anthropometric Measurements

Height was measured by the same experienced evaluator who was a Level II ISAK anthropometrist. Measurements were taken to the nearest 0.1 cm using a stadiometer (GPM, Seritex, Inc., Carlstadt, NJ, USA), according to the International Society for Advancement of Kinanthropometry (ISAK) protocol [18]. Body composition was measured using a segmental body composition analyser (hand-to-feet) (Tanita BC-545N, Tanita Corp, Tokyo, Japan). The monitor consists of an eight-electrode system, with electrodes placed on the metal foot plates and handles.

### 2.4. Statistical Analysis

Means are presented for all quantitative variables alongside standard deviations. Data normality was tested using the Shapiro-Wilk test with Lilliefors correction, and homoscedasticity was assessed using the Levene test. Following verification that data were non-normally distributed, Mann-Whitney U tests were employed for two-group comparisons. Average daily energy and macro and micronutrient intake were compared between disciplines, and consumption of each food group was compared according to sex and discipline. Data were analysed using the IBM-SPSS version 25.0 statistical programme for Windows (Armonk, NY, USA: IBM Corp). The level of significance was set at 0.05.

## 3. Results

Daily intake of energy and macronutrients according to sex and cycling discipline is shown in Table 2. No statistically significant differences were found as a function of cycling discipline in either males or females. Cyclists reported adequate energy intake (using active level of physical activity), with the exception of male XCO, who reported a slightly lower energy intake than recommended levels (3009.5 vs. 2607.9 kcal/day). Protein intake exceeded the PRI recommendation of 0.83 g/kg/day for all cycling groups: male road cyclists (1.76 g/kg/day), male XCO cyclists (1.70 g/kg/day), female road cyclists (1.65 g/kg/day), and female XCO cyclists (1.83 g/kg/day). Fat exceeded RI recommendations in females in both road (43.3%) and CXO (39.8%) cycling groups, whilst males were shown to follow recommendations for fat intake. CHO was below recommended levels in all groups: male road cyclists (43.1%), female road cyclists (36.6%), and female XCO cyclists (39.5%). The only exception was seen in male XCO cyclists (45.6%). Fibre consumption was also below recommended levels in all groups: male road cyclists (17.6 g), female road cyclists (13.2 g), and female XCO cyclists (18.3 g), with the only exception being for male XCO cyclists (32.7 g).

Table 3 describes cyclist micronutrient intake as a function of discipline and RDA/AI values. No statistically significant differences were found according to cycling discipline in either males or females. When adequacy was assessed with respect to RDA/AI (above 80%), the intake of all vitamins exceeded recommendations, with the exception of B9 in female road cyclists (77.8% RDA) and vitamin D in all groups (male road cyclists: 44.7%; male XCO cyclists: 26.7%; female road cyclists: 20.7%; female XCO cyclists: 32.7%). With regards to mineral intake, consumption exceeded RDA/AI recommendations in all groups, with the exception of iodine in male XCO cyclists (61.6%), female road cyclists (61.6%), and female XCO cyclists (58%) in addition to potassium in female road cyclists (74.6%).

Males consumed greater amounts of eggs (1.71 vs. 0.68 serving/day; *p* ˂ 0.05) and non-processed foods (2002.8 vs. 1218.7 g/day; *p* ˂ 0.05) than females. Road cyclists consumed greater amounts of fish and seafood (1.68 vs. 0.93 serving/day; *p* ˂ 0.05) and had a lower intake of coffee and tea (0.37 vs. 1.52 serving/day; *p* ˂ 0.05) than XCO cyclists (Table 4).

## 4. Discussion

The main aim of the present study was to assess the nutritional intake of elite cyclists during pre-season. The main finding was that professional cyclists in the present study consumed an unbalanced diet during this period, which was characterised by an excess of protein and fat and an insufficiency of CHO regardless of sex and cycling discipline. During a complete year, cyclists generally met consumption recommendations of the Spanish Society of Community Nutrition (SENC) for all food groups, with the exception of meat and meat products, fish, and eggs, with consumption in this case being above recommendations.

Total CHO intake was 40.77 ± 8.47% of total energy intake (TEI), with no differences according to cycling discipline. Males in both road (43.1% TEI) and XCO (45.6% TEI) disciplines reported intakes that were close to the lower reference limit (RI) for CHO (45–60% TEI), whilst females in both road (36.6% TEI) and XCO (39.5% TEI) disciplines consumed below RI levels of CHO. Specific guidelines for CHO intake [7] to enable daily recovery and meet fuel needs in cyclists undertaking a light training state prescribe an amount of 5–7 g/kg BM/d. Both males (4.5 ± 1.8 g/kg BM/d) and females (3.9 ± 1.7 g/kg of BM/d) reported intakes that were below recommended levels. A high CHO intake has previously been reported during different events such as the Tour of Spain (12.5 g/kg of BM/d) [1] and the Tour of France [5], with CHO intakes of 61% of the TEI being reported. However, no previous studies have examined CHO intake during the types of “free living” or unsupervised training periods inherent to pre-season. It is possible that during the pre-season, cyclists focus on reducing CHO intake in an attempt to maintain an appropriate body composition given that nutritionists are not on hand to orientate their diet when they are on vacation during this period.

The present study revealed overall protein intake to be higher than recommended levels in endurance athletes (1.2 to 1.4 g/kg BM/day) in all study groups [10]. Previous reports indicate that endurance athletes generally consume more protein than thought to be required [19]. High-protein diets are typical in professional athletes [20] even during pre-season. Similar trends have been shown in recreational Spanish cyclists and triathletes, with a protein intake of 1.4 ± 0.5 g/kg of BM/d being reported. This may be explained by the decline in adherence to a Mediterranean diet in Mediterranean countries over recent decades, with a shift towards Western-type diets [21] emerging, which are richer in red and processed meats.

With regards to total fat intake, the present study showed a higher fat intake than that reported in previous studies during cycling events [1,2,3]. During the competitive season, CHO provides the main fuel for cyclists, as it is essential for sustaining performance. CHO is not as important during the pre-season, leading cyclists to swap CHO for other macronutrients such as fats. In the present study, males met IR amounts, whilst intake in females was above that recommended. When engaging in moderate physical activity, it is recommended that 30% of overall energy intake is constituted by CHO, with this increasing to 35% when engaging in vigorous physical activity [22]. A possible explanation behind females consuming above-recommended amounts is that cycling is less professionalised than in the case of men, and they may be less concerned about maintaining appropriate body weight for performance during this period.

Intake of all vitamins and minerals exceeded 80% of their respective RDA or AI, with the exception of vitamin D and, in the specific case of female road cyclists, vitamin B9 and potassium. Below recommended consumption of vitamin D has previously been reported in Spanish cyclists during a racing event [1,2] and in recreational Spanish cyclists generally [23]. Although vitamin D is required for adequate calcium absorption and the promotion of bone health, Holick [24], it should be noted, emphasised that the human organism also acquires this vitamin endogenously through sun exposure.

With regards to food patterns during a complete year, professional cyclists met the daily consumption recommendations outlined by the SENC for all food groups (SENC, 2019), with the exception of meat and meat products, fish, and eggs, with intake in these cases being higher than recommendations. Cyclists in the present study met the minimum requisite of consuming two servings of vegetables and up to three servings of fruit a day. This distances them from the Spanish population, which tends to consume approximately two servings of fruit and vegetables a day [25]. The present study reported an intake of 2.62 servings a day of milk and dairy products, approximately 3.43 servings a week of pulses, and 1.10 servings a day of nuts, with this being in line with SENC recommendations. When comparing milk and dairy product intake outcomes with other Spanish populations, it can be observed that professional Spanish cyclists tend to consume more than the general Spanish population (257.2 g/day) [25] and Spanish recreational athletes (2.39 servings a day) [23]. With regards to the consumption of pulses, professional Spanish cyclists reported a higher intake than the Spanish general population (1 serving per week) [25] and Spanish recreational athletes (2.45 servings a week) [23]. Professional cyclists reported an average consumption of 1.72 servings of meat and meat products a day compared with the maximum of three servings a week recommended by the SENC. Spanish professional cyclists have reported much higher consumptions of meat and meat products than Spanish recreational athletes (2.5 serving a week) [23] and the Spanish general population, with an average daily consumption of 146 g/day [25]. Most of the meat and meat products consumed by cyclists comes from white meat. However, this high consumption of meat and meat products should be highlighted, as a high consumption of red meat and especially processed meat is associated with an increased risk of several major chronic diseases and premature mortality [26]. Spanish professional cyclists also reported a higher intake of fish and seafood consumption than that outlined by SENC guidelines (3–4 servings per week) and that consumed by the general Spanish population (88.9 g/day) and recreational Spanish athletes (0.83 servings a day). Although it is important to consume n-3 PUFA as a means to providing the body with the EPA and DHA required for optimal physiological functioning (FAO/WHO, 2008), excess protein (>2 g/kg BW/day for adults) may result in digestive, renal, and vascular abnormalities and should be avoided [27].

Ultra-processed food is related with poor diet quality and health outcomes [28] and is becoming a dominant component of diets worldwide [29]. The Spanish population reported consumption of these foods to constitute 24.4% TEI [30], whilst recreational Spanish cyclists and triathletes reported an intake of 173.6 g/d. Lower consumptions of ultra-processed foods were reported by professional Spanish cyclists regardless of sex and discipline. This may be related with the fact that the present population has better knowledge of nutrition and understands the effect of this type of food on health and performance.

A limitation of the present study is its cross-sectional design, which inhibits examination of causal relationships. A relatively small number of cyclists was also included; however, the sample included top elite cyclists. Further, the risk of measurement error is inherent to questionnaires. Nonetheless, the FFQ has previously demonstrated high validity and reliability within similar populations. Further, a well-trained professional nutritionist administered the questionnaire in order to minimize error.

## 5. Conclusions

In conclusion, Spanish cyclists followed an unbalanced diet, which was low in CHO and high in protein, with female cyclists also consuming too much fat during pre-season. Better knowledge of food guidelines in terms of servings and varieties of food is important for professional cyclists, as it may impact their health and performance. The importance of delivering nutritional education at training camps should be highlighted in order to improve dietary strategies and support training adaptations and health.

## Figures and Tables

**Table 1 nutrients-14-03695-t001:** General characteristics of cyclist.

	Male (*n* = 15)			Female (*n* = 23)		
	Road (*n* = 8)	XCO (*n* = 7)	*p*-Value	Road (*n* = 8)	XCO (*n* = 15)	*p*-Value
**Age (y)**	20.38 ± 1.92	26.43 ± 6.40	0.047	21.13 ± 2.95	20.07 ± 10.24	**0.005**
**Weight (kg)**	71.13 ± 5.84	67.8 ± 9.11	0.411	56.95 ± 5.23	56.03 ± 4.27	0.526
**Height (cm)**	183.06 ± 4.77	179.16 ± 9.14	0.309	164.91 ± 4.74	163.49 ± 5.05	0.630
**BMI (kg/m^2^)**	21.22 ± 1.66	21.03 ± 1.31	0.815	20.98 ± 1.54	20.94 ± 1.70	0.421
**Body Fat (%)**	11.37 ± 3.92	8.43 ± 2.39	0.116	20.60 ± 4.79	18.85 ± 3.06	0.106
**Muscle mass (kg)**	58.47 ± 4.51	58.96 ± 7.81	0.889	42.83 ± 2.15	42.92 ± 3.08	0.383

BMI, body mass index.

**Table 2 nutrients-14-03695-t002:** Average daily energy and macronutrients intake of cyclists during pre-season.

		Male (*n* = 15)				Female (*n* = 23)		
	RI/AR */PRIs ^†^/AI ^‡^	Road (*n* = 8)	XCO (*n* = 7)	*p*-Value	RI/AR */PRIs ^†^/AI ^‡^	Road (*n* = 8)	XCO (*n* = 15)	*p*-Value
Energy (kcal)	3009.5 *	2943.0 ± 1028.7	2607.9 ± 632.6	0.469	2412.3 *	2203.1 ± 740.3	2278.2 ± 862.0	0.837
Protein (g)		124.4 ± 35.7	115.9 ± 32.4	0.643		93.0 ± 21.8	101.1 ± 35.9	0.567
Protein (%)		17.7 ± 3.9	18.0 ± 3.6	0.887		17.6 ± 3.3	18.1 ± 3.3	0.753
Protein (g/kg)	0.83 ^†^	1.76 ± 0.51	1.70 ± 0.38	0.813	0.83 ^†^	1.65 ± 0.45	1.83 ± 0.68	0.451
CHO (g)		320.8 ± 128.3	302.7 ± 120.1	0.784		204.4 ± 85.9	224.5 ± 93.8	0.619
CHO (%)	45–60	43.1 ± 6.6	45.6 ± 12.3	0.630	45–60	36.6 ± 6.7	39.5 ± 7.4	0.343
CHO (g/kg)		4.5 ± 1.8	4.5 ± 1.9	0.979		3.5 ± 1.3	4.1 ± 1.8	0.476
Simple CHO (g)		120.5 ± 46.6	113.6 ± 37.2	0.760		84.0 ± 48.6	90.2 ± 43.8	0.765
Fibre (g)	25 ^‡^	32.7 ± 15.2	17.6 ± 12.4	0.057	25 ^‡^	13.2 ± 7.5	18.3 ± 8.7	0.174
Fat (g)		117.8 ± 51.7	95.9 ± 33.1	0.355		106.7 ± 42.0	101.8 ± 44.5	0.797
Fat (%)	20–35	35.8 ± 7.1	33.9 ± 10.1	0.682	20–35	43.3 ± 6.9	39.8 ± 6.8	0.252
SFA (g)		29.1 ± 12.4	28.2 ± 6.0	0.865		27.3 ± 12.0	28.1 ± 13.9	0.892
SFA (%)		9.0 ± 2.9	10.2 ± 3.0	0.451		10.8 ± 2.1	10.9 ± 2.6	0.971
MUFA (g)		56.5 ± 27.0	40.0 ± 17.7	0.192		52.8 ± 24.7	46.8 ± 19.0	0.527
MUFA (%)		17.1 ± 4.4	14.1 ± 5.5	0.268		21.2 ± 5.0	18.6 ± 3.7	0.157
PUFA (g)		18.5 ± 9.5	15.9 ± 8.5	0.587		16.6 ± 7.7	16.9 ± 11.0	0.949
PUFA (%)		5.5 ± 1.0	5.4 ± 2.3	0.960		7.0 ± 3.2	6.4 ± 2.1	0.619
Cholesterol (mg)		474.1 ± 146.4	347.0 ± 152.1	0.123		370.4 ± 114.6	333.9 ± 137.6	0.529

CHO, carbohydrates; SFA, saturated fatty acid; MUFA, monounsaturated fatty acid; PUFA, polyunsaturated fatty acid; RI:, reference intake; AR *, average requirement; PRIs ^†^, population reference intake; AI ^‡^, adequate intake. Using level of PA: active.

**Table 3 nutrients-14-03695-t003:** Average daily vitamins and mineral intake of cyclists during pre-season.

		Male (*n* = 15)				Female (*n* = 23)		
	RDA/AI *	Road (*n* = 8) (%RDA/AI)	XCO (*n* = 7) (%RDA/AI)	*p*-Value	RDA/AI *	Road (*n* = 8) (%RDA/AI)	XCO (*n* = 15) (%RDA/AI)	*p*-Value
Vitamins								
A (µg)	900	1646.8 ± 924.6 (183.0)	966.1 ± 341.4 (107.3)	0.084	700	775.6 ± 372.0 (110.8)	906.7 ± 462.7 (129.5)	0.499
B1 (mg)	1.2	2.6 ± 1.4 (216.7)	2.2 ± 0.7 (183.3)	0.473	1.1	2.1 ± 1.4 (190.9)	1.8 ± 0.8 (163.6)	0.483
B2 (mg)	1.3	3.1 ± 1.5 (238.5)	2.7 ± 0.6 (207.7)	0.501	1.1	2.7 ± 1.4 (245.5)	2.1 ± 0.9 (190.9)	0.222
B3 (mg)	16	53.2 ± 14.4 (332.5)	48.6 ± 7.6 (303.8)	0.446	14	40.0 ± 9.5 (285.7)	45.1 ± 19.1 (322.1)	0.487
B6 (mg)	1.3	4.2 ± 2.1 (323.1)	3.3 ± 0.8 (253.8)	0.336	1.3	2.5 ± 0.5 (192.3)	3.1 ± 1.7 (238.5)	0.390
B9 (µg)	400	660.4 ± 352.6 (165.1)	376.6 ± 158.9 (94.2)	0.072	400	311.0 ± 80.8 (77.8)	381.1 ± 181.1 (95.3)	0.314
B12 (µg)	2.4	9.4 ± 4.8 (391.7)	5.7 ± 3.2 (237.5)	0.107	2.4	7.2 ± 4.0 (300)	10.5 ± 9.2 (437.5)	0.354
C (mg)	90	208.0 ± 113.5 (231.1)	139.0 ± 69.7 (154.4)	0.187	75	91.4 ± 29.6 (121.9)	130.7 ± 92.7 (174.3)	0.260
D (µg)	15 *	6.7 ± 5.0 (44.7)	4.0 ± 4.0 (26.7)	0.280	15 *	3.1 ± 1.1 (20.7)	4.9 ± 4.1 (32.7)	0.189
E (mg)	15	20.5 ± 11.9 (136.7)	14.4 ± 7.3 (96)	0.257	15	23.0 ± 18.0 (153.3)	15.9 ± 7.7 (106)	0.192
**Minerals**								
Calcium	1000 *	1110.8 ± 398.4 (111.1)	1020.9 ± 357.3 (102.1)	0.655	1000	982.0 ± 623.7 (98.2)	904.9 ± 433.3 (90.5)	0.731
Iron	8	26.3 ± 13.3 (328.8)	21.6 ± 7.9 (270)	0.429	18	17.3 ± 4.3 (96.1)	17.3 ± 7.4 (96.1)	0.997
Zn	11	14.4 ± 4.0 (130.9)	13.4 ± 4.1 (121.8)	0.641	8	10.8 ± 3.4 (135)	11.1 ± 3.8 (138.8)	0.879
Mg	400	564.9 ± 225.9 (141.2)	470.7 ± 182.1 (117.7)	0.395	310	391.9 ± 153.3 (126.4)	408.5 ± 206.6 (131.8)	0.844
Na	1500 *	2787.5 ± 1007.4 (185.8)	4025.7 ± 2366.2 (268.4)	0.199	1500 *	2847.3 ± 1511.9 (189.8)	3302.0 ± 46.7 (220.1)	0.464
K	4700 *	5131.5 ± 1490.8 (109.2)	3923.0 ± 1487.8(83.5)	0.141	4700 *	3508.6 ± 1089.6 (74.6)	3831.9 ± 1968.3 (81.5)	0.673
Mn	2.3 *	6.4 ± 4.2 (278.3)	5.2 ± 2.9 (226.1)	0.521	1.8 *	3.2 ± 1.6 (177.8)	3.9 ± 2.0 (216.7)	0.408
P	700	2042.1 ± 622.0 (291.7)	1859.4 ± 459.2 (265.6)	0.534	700	1623.3 ± 531.1 (231.9)	1735.6 ± 685.4 (247.9)	0.692

RDA, Recommended Dietary Allowances in ordinary type; AI *, Adequate Intakes in ordinary type followed by an asterisk. An RDA is the average daily dietary intake level that is recommended as being sufficient to meet the nutrient requirements of nearly all (97–98 percent) healthy individuals in a group. The AI is believed to cover the needs of all healthy individuals in the groups, but lack of data or uncertainty in the data prevent specific conclusions on the proportion of the sample achieving AI from being reported.

**Table 4 nutrients-14-03695-t004:** Consumption of each food group according to sex and discipline during the complete season.

	Overall (*n* = 38)	Male (*n*= 15)	Female (*n*= 23)	*p*-Value	Road (*n*= 16)	XCO (*n*= 22)	*p*-Value
Food groups (serving/day)							
Milk and dairy products	2.62 ± 1.64	2.78 ± 2.04	2.52 ± 1.36	0.685	2.60 ± 2.03	2.65 ± 1.35	0.944
Eggs	1.09 ± 0.98	1.71 ±1.22	0.68 ± 0.51	0.011	1.35 ± 0.91	0.89 ± 1.02	0.192
Meat and meat products	1.72 ± 0.92	1.81 ± 0.86	1.65 ± 1.01	0.822	1.83 ± 1.22	1.64 ± 0.74	0.788
Fish and seafood	1.25 ± 0.82	1.48 ± 1.10	1.10 ± 0.55	0.258	1.68 ± 0.94	0.93 ± 0.55	0.007
Vegetables	3.31 ± 2.24	3.86 ± 1.88	2.96 ±2.42	0.267	3.68 ± 2.31	3.03 ± 2.20	0.415
Pulses	0.49 ± 0.57	0.60 ± 0.81	0.41 ± 0.34	0.384	0.47 ± 0.75	0.51 ± 0.37	0.850
Fruits	3.68 ± 2.05	4.36 ± 1.51	3.16 ± 2.29	0.115	3.47 ± 2.09	3.03 ± 2.20	0.611
Nuts	1.09 ± 0.92	1.32 ± 0.94	0.93 ± 0.91	0.280	1.10 ± 0.98	1.08 ± 0.90	0.953
Pasta, potatoes, rice	1.15 ± 0.76	1.26 ± 0.79	1.08 ± 0.76	0.518	1.15 ± 0.88	1.15 ± 0.68	0.995
Refined cereals	1.00 ± 0.89	1.32 ± 1.11	0.78 ± 0.64	0.112	0.96 ± 1.16	1.04 ± 0.58	0.802
Whole cereals	0.46 ± 0.58	0.53 ± 0.80	0.41 ± 0.37	0.585	0.43 ± 0.72	0.49 ± 0.43	0.795
Olive oil	1.83 ± 1.31	2.21 ± 1.77	1.59 ± 0.87	0.300	2.29 ± 1.59	1.37 ± 0.75	0.062
Other oils and fats	0.24 ± 0.61	0.16 ± 0.36	0.29 ± 0.74	0.589	0.27 ±0.81	0.21 ± 0.34	0.813
Baked goods and pastries	0.37 ± 0.055	0.20 ± 0.15	0.49 ± 0.69	0.123	0.34 ± 0.67	0.40 ± 0.43	0.813
Sugar and sweets	1.10 ± 1.33	1.54 ± 1.80	0.83 ± 0.91	0.192	0.65 ± 0.88	1.62 ± 1.61	0.064
Beer/wine	0.07 ± 0.15	0.02 ± 0.04	0.11 ± 0.19	0.105	0.09 ± 0.17	0.06 ± 0.13	0.652
Alcoholic beverage	0.00 ± 0.01	0.00 ± 0.00	0.01 ± 0.02	0.410	0.01 ± 0.02	0.00 ± 0.00	0.410
Soft drinks	0.10 ± 0.21	0.14 ± 0.29	0.06 ±0.12	.385	0.02 ± 0.05	0.20 ± 0.29	0.081
Coffee/tea	0.85 ± 1.21	0.59 ± 0.79	1.04 ± 1.44	0.388	0.37 ± 0.58	1.52 ± 1.55	0.047
Unprocessed food (g/d)	1548.0 ± 901.9	2002.8 ± 1023.3	1218.7 ± 696.8	0.025	1677.4 ± 1176.0	1417.2 ± 671.1	0.466
Unprocessed food (g/d/kg)	24.8 ± 13.3	29.0 ± 13.4	22.0 ± 12.8	0.151	25.7 ± 15.1	24.1 ± 12.1	0.750
Ultra-processed food (g/d)	72.2 ± 88.6	52.3 ± 60.8	85.0 ± 102.2	0.699	58.7 ± 64.4	82.1 ± 103.5	0.432
Ultra-processed food (g/d/kg)	1.2 ± 1.5	1.2 ± 2.0	1.2 ± 1.1	0.953	1.0 ± 1.1	1.4 ± 1.7	0.351

g/d, grams per day.

## Data Availability

The dataset generated and analyzed during the current study is available from the corresponding author on reasonable request.

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
