# Peer review of "Nutritional Habits of Professional Cyclists during Pre-Season"

_nutrients, 2022, doi:10.3390/nu14183695_

Round 1

Reviewer 1 Report

Overall, a fascinating, well-written study examined the dietary intake of professional cyclists during pre-season and during a complete season. 

The introduction is well written.

Results

The authors found that fat exceeded RI recommendations in 141 females in both roads (43.3%) and CXO (39.8%) cycling groups, while males were shown to follow recommendations for fat intake. Since the authors assessed daily nutrient intake using a 72-h dietary recall interview, I would suggest examining whether there is a relation between fat consumption and the menstrual cycle in women. 

Also, I would suggest that the authors conduct a sample size and power analysis.

Author Response

Reviewer #1

[comments]

Overall, a fascinating, well-written study examined the dietary intake of professional cyclists during pre-season and during a complete season. The introduction is well written.

[response]

Many thanks, we appreciate the reviewer’s comments. We have tried to clarify details of the study according to the recommendations provided.

 [comments]

Results

The authors found that fat exceeded RI recommendations in 141 females in both roads (43.3%) and CXO (39.8%) cycling groups, while males were shown to follow recommendations for fat intake. Since the authors assessed daily nutrient intake using a 72-h dietary recall interview, I would suggest examining whether there is a relation between fat consumption and the menstrual cycle in women. 

Also, I would suggest that the authors conduct a sample size and power analysis.

[response]

Thank you for your recommendation. It could be interesting to examine the relationship between daily nutrient intake and menstrual cycle in women. Unfortunately, the study was cross-sectional in nature and no information was collected with regards to the menstrual cycle.

With regards to the sample size and power analysis, the sample included all cyclists participating in a national pre-season training camp and so by its very nature is representative of elite road and mountain bike cyclists in Spain.

Reviewer 2 Report

The manuscript nutrients-1879534 named “Nutritional habits of professional cyclists during pre-season” assessed the eating habits of 15 male and 23 female elite cyclists during pre and complete season.  The daily nutrient intake was assessed using a 72-h dietary recall interview considering intake over the previous three days, including one weekend day to capture weekly variations on weekdays. It was observed a greater consumption of proteins for all cycling groups and fat for female cyclists, while carbohydrates intake was below recommendations in all groups. The present study is interesting, and it is well-writer, however, presents a description of data based on nutritionists’ interviews. Some points must be improved, and general comments were detailed below.

Other keywords may be added, such as eating habits, cyclists. 

The statistical analysis was performed, but what groups were compared? Male and female? Or Road and XCO? Some data was described in percentages, but this point needs a better description. 

Spearman correlations analysis could also be performed. 

In the Results section, the authors need to detail the presented data for all table titles, and specifically in table 2 some symbols (*, ┼) are used, but no explanation was inserted. Does represent a statistical significance? So, is pvalue lower than 0.05? Or correct pvalue was inserted in the table and no significant differences were detected.  The authors must clarify these points.

In the discussion section, the author described the results again and no comparison with other athletes or sportiest is performed. No study limitations were inserted, and this is an important point in a scientific study.

Author Response

Reviewer #1

[comments]

The manuscript nutrients-1879534 named "Nutritional habits of professional cyclists during pre-season" assessed the eating habits of 15 male and 23 female elite cyclists during pre and complete season.  The daily nutrient intake was assessed using a 72-h dietary recall interview considering intake over the previous three days, including one weekend day to capture weekly variations on weekdays. It was observed a greater consumption of proteins for all cycling groups and fat for female cyclists, while carbohydrates intake was below recommendations in all groups. The present study is interesting, and it is well-writer, however, presents a description of data based on nutritionists' interviews. Some points must be improved, and general comments were detailed below.

[response]

Many thanks, we appreciate the reviewer’s comments. We have tried to clarify details of the study according to the recommendations provided.

 [comments]

Other keywords may be added, such as eating habits, cyclists.

[response]

Thank you for your recommendation. We have added these keywords.

[comments]

The statistical analysis was performed, but what groups were compared? Male and female? Or Road and XCO? Some data was described in percentages, but this point needs a better description.

[response]

Thank for the comment. The groups being compared are described in the statistical analysis section. Only absolute values were compared. Percentages in tables represent the percentage of RDA or AI recommendations, as described in the table.

[comments]

Spearman correlations analysis could also be performed.

[response]

A correlation analysis was conducted prior to the analysis presented in the paper. No relevant findings were produced and so it was decided to omit presentation of these results in the table in order to aid reader understanding.

[comments]

In the Results section, the authors need to detail the presented data for all table titles, and specifically in table 2 some symbols (*, ┼) are used, but no explanation was inserted. Does represent a statistical significance? So, is pvalue lower than 0.05? Or correct pvalue was inserted in the table and no significant differences were detected.  The authors must clarify these points.

[response]

Thank you for your comment. The symbols used relate to the different types of recommendations used for analysis (reference intake, average requirement, population reference intake or adequate intake). This information has also been added to the foot of the table in order to aid understanding. All p values are reported where relevant.

[comments]

In the discussion section, the author described the results again and no comparison with other athletes or sportiest is performed. No study limitations were inserted, and this is an important point in a scientific study.

[response]

As indicated in the manuscript, to our knowledge no data evaluating the nutritional intake of elite cyclists during pre-season are available, with information on energy and nutrient requirements during this period also lacking. In the discussion, outcomes are compared with general population and non-professional cyclists from the same country as differences are likely to be of interest. Limitations have now been added to the text.